# Strategies for Cost-Effectiveness in Sustainable Quality Healthcare Delivery in Emerging Economies: The Case of Healthcare Professionals Development in South Africa

**DOI:** 10.3390/healthcare13010036

**Published:** 2024-12-28

**Authors:** Francis Ikechukwu Igbo, Kenneth Gossett, Deborah Nattress

**Affiliations:** Department of Business Administration, College of Management and Human Potential, Walden University, Washington, MN 55401, USA; kenneth.gossett@mail.waldenu.edu (K.G.); deborah.nattress@mail.waldenu.edu (D.N.)

**Keywords:** cost-effectiveness, quality healthcare, sustainable healthcare, emerging markets, digital ecosystem

## Abstract

**Purpose**: This paper suggests strategies for professionals’ continuous development in healthcare institutions to ensure quality and sustainable healthcare delivery in a cost-effective way. Background: Healthcare services are increasingly becoming expensive, and receiving quality service is often difficult. This plunges practices and healthcare institutions into the sphere of the population’s mistrust. They believe in the degradation of quality due to individual experiences, with the direct corollary of reduction in life expectancy in some areas. We therefore propose strategies for enhancing the quality of those individuals for a sustainable healthcare delivery in an emerging market economy, based on the case in South Africa. **Method**: Cost-effectiveness analysis is chosen for the purpose of non-monetary analysis, and we make use of the qualitative methodology to explore, in detail, the need of strategies to improve healthcare delivery. **Results**: We propose a suitable digital ecosystem for the professionals’ development, and we evaluate the proposed methodology and the challenges that come from its realization. **Conclusions**: We find that the application of these strategies yields efficiency and quality, which, when applied continuously, yields sustainability.

## 1. Introduction

Advancements in technology have led to the profound transformation of the entire healthcare delivery system, from patient and professional requirements up to an adaptive complex regulatory system. The complex adaptive regulatory system regulates interdependencies, interconnectivity, and interactions among different components of the healthcare system. Within this emerging landscape, the South African government has recently adopted a National Healthcare Insurance (NHI) that tends to shape the path of the regulatory system, with the aim to address issues related to sustainable quality healthcare delivery to patients. The NHI is mostly accessible by all South Africans from the healthcare system. The South African healthcare system is already overloaded by issues: management issues, an insufficiency of resources, quasi-inexistant resource in rural areas, and significant patient volume in urban hospitals due to increases in urban population and increases in health issues that come from overcrowded places. There will be a need for healthcare staff to accommodate the weight that the NHI will bring to a system already burdened with management and professional scarcity problems. The study carried out in [1] weighed management development among the causes of healthcare system degradation, to which we can add the mistrust of healthcare workers from the South African populations in terms of quality. The authors of [2] stated that a sustainable healthcare delivery system is achieved when healthcare practitioners deliver high-quality care and improved public health without exhausting available resources or causing severe ecological damage. This signifies that healthcare professionals play a crucial role in ensuring the system’s delivery quality. To effectively accommodate such an insurance revolution and improve the system in terms of administration and sustainable healthcare delivery, the healthcare system must prioritize, among other factors, the training and development of professionals. However, with current bad or miserable economic situation, this requires a cost-effective strategy proposition. This study conducted a related qualitative study based on cost-effectiveness analysis. Cost-effectiveness analysis is a theoretical instrument used to compare two outcomes based on the relative costs to evaluate which of the two provides the best opportunities for success and quality maximization [3]. The analysis is used in this paper to explore strategies aiming to develop healthcare professionals for sustainable quality delivery. Such a development includes a wide array of critical areas, including, but not limited to, the enhancement of continuous professional development (CPD) that aligns with the global strategy [4].

This global strategy is intended to create a common understanding for all World Health Organization (WHO) Member States of the importance of digital health solutions, and an approach to putting in place an inter-operable digital health ecosystem. A digital ecosystem is a digital inter-operable information technology and telecommunication infrastructure that is mainly utilized by the healthcare community across all types of care operations, namely, healthcare providers, health service providers, and patients, as well as public health authorities, universities, and research institutions. According to the WHO, the inter-operability of the digital health ecosystem should enable the seamless and secure exchange of health data between users, healthcare providers, health system managers, and health data services. This paper suggests insights within this seamless inter-operability as described by the WHO’s strategy based on professionals’ development.

Professional development initiatives are not new in health systems; healthcare continuous professional development (CPD) has already been reported in [5,6,7,8], as well continuous medical education (CME) [5]. CPD is a continuous process in which the quality of health professions is consistently and continuously enhanced through learning commitment from all health practitioners [9]. CME, according to the American Medical Association (AMA), is made up of “educational activities which serve to maintain, develop, or increase the knowledge, skills, and professional performance and relationships that a physician uses to provide services for patients, the public, or the profession” [10]. Although CPD and CME are frequently used interchangeably, most of the literature now defines CME as an ingredient of CPD [11]; hence, our focus is on CPD. To date, CPD in South Africa is a purposeful statutory process meant for practitioner registration with a body called the South African National Council (SANC). Once registered, the practitioner commits to and engages in a range of learning activities to maintain and improve their knowledge, skills, attitude, and professional integrity to keep up to date with new science, innovations, and healthcare developments [12]. The system put in place only mentions activities that generate points for practitioners, but it does effectively follow up with these activities. In this regard, this paper suggests an enhanced CPD strategy that applies to the whole healthcare workforce, which is spread across a wide range of different health sectors and different health organizations and settings, acquired through a digital health ecosystem. This strategy is derived from different opinions and expectations from healthcare professionals obtained through surveys and interviews collected at different healthcare institutions and analyzed using cost-effectiveness tools.

The contribution of this paper is to define a CPD strategy that enables the healthcare system to implement a digital training and development platform that accompanies the NHI, with the aim to satisfy the forthcoming demand in terms of healthcare professionals and the sine qua none condition for a sustainable quality healthcare delivery in South Africa. In doing so, we conducted surveys and question interviews analyzed through a cost-effective study, based on some professionals from healthcare institutions in Gauteng province, South Africa.

This paper is organized as follows: Section 2 reviews the literature related to healthcare cost-effectiveness, Section 3 presents management issues of the healthcare system in South Africa, and Section 4 analyzes the cost-effectiveness to determine how much professional development is needed in South Africa. Section 5 derives the strategies through findings from the data collection, Section 6 proposes the implementation of these strategies, Section 7 evaluates the implementation through discussions, and Section 8 concludes the paper.

## 2. Literature Review

The healthcare system has undoubtfully taken a path toward its revolution, driven significantly by advancements in technology [13]. Thus revolution requires healthcare organizations to commit and remain committed to enhancing the skills and knowledge of their healthcare professionals through continuous investments in technology-based training and education [14,15]. Committing professionals to knowledge management, training, and professional development using tools that aim to improve operational efficiency becomes crucial for the growth and success of healthcare organizations [16]. Many entities in this regard have already established comprehensive policies and procedures to ensure professionals’ continuous development [17].

According to [18], CPD is a learning experience that yields the development and improvement of professional practice based on one’s strengths, while developing professionals to fill their capability gaps. It is therefore important to set up such opportunities, tools, and resources that include CPD into the flow of healthcare institution work life according to The Chartered Institute of Personnel and Development presented in 2024 [19], thus developing professional practice with the following benefits: keep skills and knowledge up to date; prepare professionals for greater responsibilities; boost one’s confidence; help individuals to become more creative in tackling new challenges; enable individuals to make better decisions; and help individuals take their career further [20]. In another parallel direction and according to [21], CDP allows us not only to update knowledge and skills, enhance carrier development, or improve client services and experiences but also to practice safely and attain social advantages [22,23]. This paper suggests parallel strategies to the healthcare system in emerging countries in general and South Africa in particular based on a cost-effectiveness analysis.

A cost-effectiveness analysis framework refers to comparing the relative costs of achieving the same outcome using different activities [24]. It is a theoretical tool that can be used to compare two outcomes based on the relative costs with the aim of evaluating which of the two provides the best opportunities for success and quality maximization [25]. This cost-effectiveness comparative tool is therefore used in this study to investigate the need of professionals’ development and its optimization at the dawn of the technological revolution of the health sector for better management and quality delivery in South Africa.

## 3. Healthcare System Management in South Africa

The South African healthcare system is made up of basic primary healthcare and highly specialized healthcare provided by both public and private healthcare services. Public healthcare provides services to most of the population (about 84%). However, public health service is largely underfunded, short-staffed (see Table 1), and deteriorated with only approximately 40% of government contribution in all expenditures in the public healthcare sector, pressuring the service delivery sector [26]. Consequently, only about 54% of the population come out of the country’s public healthcare facilities satisfied [26], creating mistrust of the healthcare system in general, and of healthcare workers in particular. Contrary to the private healthcare sector, which is mostly accessible by middle- and high-income earners essentially covered by private medical insurance providers, the public healthcare sector experiences poor management and waning infrastructures [27] and poor quality of healthcare delivery, especially in the poor and rural parts of the country [26]. In contrast, while life expectancy in South Africa has increased from 53 years in 2011 to 58.5 years in 2014, and to around 65 years in 2019, health issues in South Africa were worsened by the COVID-19 pandemic and other public health challenges like HIV and tuberculosis (TB). In considering the differences in race, socioeconomic status, and rural–urban differentials, in addition to increased HIV and TB prevalence, South Africa’s apartheid past still defines the healthcare service and resource inequities in the country [28]. All these challenges have put a burden on the government to fulfill the funding and maintenance of the healthcare system, especially in the public healthcare sector. Furthermore, poor and rural inhabitants, which are mostly black South Africans, experience barriers to accessible quality healthcare [29], receive a lower rate of healthcare service, and receive lesser benefits from their care [30], as the burden of ill health is more on them due to their socio-economic status [31]. It is in this bleak context, compared to that in BRICS countries, that we suggest uniform sustainable quality healthcare for all South Africans, which necessitates health professionals at all levels, in all places, with efficacy and efficiency, to continuously deliver quality service.

In addition to the system’s poor-quality delivery, as mentioned above, Table 1 shows the number of healthcare professionals in South Africa for an estimated population of 55 million people. Accordingly, not only is this number too small, particularly with the NHI, which will open access to the system to everyone, but there is also an urgent need for staff to avoid a system collapse like during COVID-19. The South African government, through a task team [31], already implemented mechanisms to develop a health workforce that enables an efficient and effective health system. The related CPD aligns with the Sustainable Development Goals (SDGs); attaining these goals results in improved population health and responsiveness to patient and community expectations and ensures financial risk protection. Unfortunately, the key targets of these goals, from SDG3 to SDG 8, which are part of the United Health System (UHS) common to all emerging countries, though inclusive, do not reach the required digitalization. Hence, we propose a digital CPD and a cost-effectiveness analysis to define the required strategies.

CPD in South Africa, as mentioned earlier, is handled by the bodies known as the SANC [11] and Allied Health Professions Council of South Africa (AHPCSA) [32]. The SANC is a statutory process meant for practitioners’ registration that allocates points in accordance with the continuing education units to registered healthcare professionals, while the AHPCSA defines the continuing education units, keeps individuals’ training records, and approves accreditations. But both bodies conduct their responsibilities manually, which sometimes results in biased trends and errors. In this paper, we suggest a more automated approach from the registration to the recording, through accreditations that not only map South Africa to the world but also remove biased behavior and human errors.

## 4. Cost-Effectiveness Analysis: Study Case in South Africa

Since the early 1990s, countries like the USA have made use of cost-effectiveness analysis in healthcare as a tool for determining how to prioritize resource allocation toward healthcare funding [33]. Cost-effectiveness analysis, which to date has been used in finance, has become an evaluation tool in medical research because of its inappropriate application in monetizing healthcare results and its ability to compare the relative costs and outcomes of different healthcare delivery possibilities. In this study, we made use of the theoretical framework of cost-effectiveness analysis to hypothesize healthcare professionals ‘development in terms of cost-effectiveness in developing them and their delivery of quality, and of healthcare sustainability incurred in the system. Therefore, we explore strategies that healthcare institutions can use to develop cost-effective and sustainable quality healthcare delivery through staff development. Thereafter, healthcare institutions as well as practitioners must endeavor to improve their skills and knowledge without compromising the institution’s tasks, which is the logical consequence of delivering quality healthcare to patients.

### 4.1. Research Method

The research method in this study uses a qualitative case study to identify approaches that health institutions can use to develop their professionals that ensures quality healthcare delivery and profitability to the institution as well as to the population. Cost-effectiveness and its conceptual framework method were chosen for this purpose. The objectives were to select a sample within the healthcare fraternity, collect data, analyze the data, and decrypt the best way that maximizes each outcome based on quality delivery. Data were collected through surveys and interviews as per primary research requirements and examined through focus group discussions. Accordingly, we conducted online interviews using virtual meeting platforms (i.e., Zoom, Teams, or Google Meet) and telephone conversations to provide variety and convenience for the different participants, thereby ensuring participants’ ability and willingness to complete the interview process as suggested in [34]. The surveys were conducted based on the Likert scale because their estimates were population values, which can be erroneous, and because they required tools like relative standard deviation evaluation; they were only used for the validity of our samples, while the interviews needed an extra focus group discussion for qualitative information extraction. Therefore, the following sampling and data collection were conducted.

### 4.2. Sample Selection

The sample selection considered the suitability of the respondents to the purpose of this study [35], as well as their predisposition in applying research questions and developing the theoretical and/or conceptual framework that complies with this research study [36]. Therefore, our population was selected as required for qualitative analysis, based only from their relevance to the research questions [37]. Based on this criterion, our study population essentially consisted of health professionals within Gauteng province in South Africa, taking advantage of their characteristics and the density of the healthcare system in the province. Therefore, three criteria were considered in selecting these research participants, in accordance with [38]: demographic characteristics, the size of their healthcare facility, and their geographical location. We used a bias method as the sampling technique, and questionnaires were distributed to healthcare workers in Gauteng prior to organizing interviews. The reason behind the delay was to use the survey to validate the sample, while avoiding any quantitative attempt. Sample validation means that we wanted to make sure of the quality of the source before starting the interview process. The first aim was to collect data from fifty (50) respondents divided into five (5) groups of ten (10), which also constituted our focus groups. All respondents completed their survey questionnaire and were consequently booked for an interview, and it is the interviews that collected the data used in this study.

In the technique applied, we performed the following:-We started with a sample with a small number of participants in each group, and carried on increasing this number, while observing variation in the data during data collection.-We carried on in the same way according to the conceptual framework while interviewing more participants-We stopped when the responses become repetitive, which was the saturation point. The saturation point was reached when one observes the same answer for the third time consecutively moving forward [37,39]. For example, when one receives three respondents consecutively saying online flexible time training. We know that a flexible timetable is the saturation point mode of training.

The framework of this sampling is given in Figure 1 [24].

Prior to data collection, our first task in this process was to reveal the profiles of our samples, namely, the individual and sector profiles. An individual profile is divided into five categories: general practitioners, nurses, emergency medical services, social workers, and pharmacy professionals. And the “sector” profile that characterizes the rubric is mainly divided into public and private sectors. In each group of five, there are two (2) newcomers in the business, two (2) experienced members, and one (1) administrator. Our major intention was to prove that, descriptively, our samples fit with the sampling technique mentioned above in the conceptual framework. These samples were integrated as shown in Table 2 below.

### 4.3. Data Collection

According to [40], the most used tools in data collection are questionnaires/surveys, interviews, observations, and focus group discussions. A primary data source (interviews) was adopted for data collection in this analysis using interviews, as well as a secondary data source based on policy documents, gazettes, and records. Most of the secondary source can be found in [12] and in the South African Government Gazette [32]. We also considered published documents on healthcare sustainability. Primary data give exclusive insights into information to explore or generate new theories, something the secondary data cannot provide. Policy documents, gazettes, and records such as the Interministerial Team of [31] informed us in terms of applied policies and plans in quantitative and qualitative ways. As a qualitative case study, recorded raw data were reproduced from the interviews conducted and substantiated with any notes taken during the interview process. The interview questions were designed in a way that allowed us to identify the questions that better help us determine strategies of cost effectiveness in sustainable healthcare delivery systems (see Table 3). The summarized or structured data were sent back to the participants to ensure consistency and the effective representation of their opinions. The data, free of inconsistencies, were coded further into sub-concepts, main concepts, and, finally, research outcomes, while direct codes and outcomes were developed from the secondary data sources [32]. This process, known as data validation, is intended to check the accuracy of the data and the quality of the source for data representation. In this coding process, data were systematically categorized from extracts of the conducted interviews and focus groups and put into themes that constitute our structural analysis for cost-effectiveness. All these steps are summarized in Figure 2, and the resulting findings from [41]. Though we are satisfied with the data validation process, it is worth mentioning the limits of the process since it is biased and we cannot control the technology used on the online platform used for interviews. However, as we continued forward, our coded data were recorded.

Using the coded data from the interview questions and focus groups, we applied the cost-effectiveness analysis framework to explicitly take a sectorial perspective, where the costs and effectiveness of all possible respondent interventions (i.e., healthcare professionals’ development) were compared to select the mix that sustainably maximizes healthcare quality for a given set of human resource constraints. Cost-effectiveness has three key concepts underlying the theory, namely, mode, quality, and cost. Throughout the study, we supported the assumption that the mode, quality, and cost of healthcare professionals’ development would affect the sustainability of quality healthcare delivery. This has resulted in the data analysis we conduct in the next section.

### 4.4. Data Analysis

We coded and recorded accurate data, now let us analyze it. We therefore make use of descriptive and diagnostic analytics based on grounded theory. Descriptive analytics allow us to look at the patterns or structure obtained prior to going through the decision process. Diagnostic analytics allow us to identify the outcomes that contribute to our strategies. In this latter stage, four major outcomes in compliance with the patterns from interviews were found. These four outcomes were derived in such a way that they integrate UHC 3 to UHC 8 of the SDG 2030 defined by the South African Government [31], and they are as follows:Outcome 1: E- and online learning:
-Justification: All five participants expressed the importance of having an online and/or e-learning program available to healthcare professionals. They all agreed that continuous training will improve the quality of the South African healthcare system because patients will be attended by learned professionals. One stated (source#7 group 2) the following: “the routine kills minds, but with a CPD, the renewed mind will provide new energy and better results”. Another point for the justification of the digitalization of the process is the manual paperwork experienced today. According to a policy document, “Individual CPD Activity Record that shall be held by the individual practitioner as a record of every learning activity attended or completed” (Source Gazette [32]), and this record must be presented “for compliance verification” (Source Gazette [32]) every time an individual enters an evaluation process.-Prospect: Content: Respondents further suggested that healthcare institutions must establish a continuous development program with educational and training institutions that can offer credits, certification, or institutionalized curriculum pathways through online and e-learning. “Having only AHPCSA for accreditations may limit the professionals for working abroad” (Source#5 group 1). This entails identifying or instituting educational and training programs at various educational settings for healthcare professionals, social workers, and individuals wanting to embrace medical careers.-Operation: Training is offered seamlessly through an online platform, suggested in accordance with [42,43], as well as through evaluation (“We must be evaluated online without bias”) (Source#2 group 1), suggesting a less human intervention in the process.
Outcome 2: Tri-partnership:
-Definition: This is inter-operation between three institutions, namely academic/training, state department, and healthcare institutions. Partners include universities and colleges; hospitals, emergency medicine agencies, and home healthcare agencies; and the government department and regulatory structures. It forms a tri-sector partnership [44].-Justification: Three of the top participants suggested coordinated action in training and job demands, which leads to a need for a better partnership at the decision-making level (Source 4, Group 4: “the political will as well as the one of healthcare managers is crucial here”).-Prospect: Therefore, efforts must made to establish a partnership among healthcare institutions, academic and training institutions, and government health and high education departments, particularly with the advent of the NHI. Health institutions assess their staff according to a predetermined training and development plan; academic and training institutions implement tailored programs in collaboration with research institutions regarding new results and their applications; and state departments encourage individuals to commit to the program through a competitive examination admission.
Outcome 3: Transformation outreach:
-Definition: It refers to an inclusive transformation toward social cohesion that is extended to all levels of South African society, including people of different races and income groups, and even offenders of society.-Justification: A social worker among the five participants believes that it is an opportunity for the healthcare system to undergo transformation (Source 5 Group 1: “With-out transformation in the South African society, nothing will yield good results”), which has been agreed on in focus groups like those in [45].-Prospect: Transformation intends to drive Africans, Europeans, and Asians to find belonging in the country’s healthcare system. Therefore, the outreach of the training must consider the requirements of South African society’s transformation. It does not exclude self-funded individuals from accessing the programs, or individuals from poor backgrounds or from correctional services with good records.-Operation: The admitted candidates are selected after successfully passing an inclusive competitive entrance examination organized by the Department of Health in accordance with the job demand, the National Students Financial Aid Scheme (NSFAS) conditions, and the transformation requirements. The selection of a program can be conducted in advance by the candidate, but mostly by the performance exhibited during the competitive entrance examination. This competitive entrance examination sets a direction toward healthcare institutions, human resources, and job-placement agencies to better understand the challenges faced by South African populations in terms of transformation and the strategies to address institutional or implicit biases in hiring. According to a pharmacist, the government must provide additional support to the newly promoted that are willing to be independent, instead of focusing on the hiring process only (Government instead of putting eyes on our revenue, should be encouraging us in doing better) (Source#2, group 5).
Outcome 4: Programs:
-Justification: Though a participant did mention an exchange forum for specialists and another general practitioners for collaboration, which according to the majority can significantly improve interventions and cure protocols (Source 2 Group 4: “cure proto-cols like what happened during COVID-19 are necessary for big health issues”), five participants proposed that programs should focus on practices that will be highly in demand mostly at the primary care level, with the advent of the NHI.-Prospect: Among the practices, we can cite the following: emergency medical services, hospital- and clinic-based workforces, tailored healthcare professional training, healthcare management programs, and broader integrated training programs, including home care and management support.


The program also most often adapts to the need of a healthcare institution (“The system that provides a short program adapted to the current need”) (Source#5, group 1).

There is no continuing development program without continuing evaluation. This evaluation must be out of human hands, which are sometimes bias (“We must be evaluated online without bias”) (Source#2 group 1), compared to the current evaluation with the SANC assigning points (“How the 15 points are given we don’t know”) (Source#1, group 2).

Outcomes 1–4 represent the strategies obtained from the analysis of the data survey collected from questionnaires/surveys and interviews, and they can be represented as the words summarized in Figure 3, and as strategies, which, through their implementations suggested in the next section, provide more insights into a cost-effective and sustainable quality healthcare system.

## 5. Proposed Implementation of Professionals’ Continuous Development Strategies

The outcomes above, denominated as strategies, emerged from the doctor’s interview outcomes meant to improve the quality of healthcare professionals and consequently provide sustainable quality delivery healthcare to all South Africa. They are presented in Figure 4 with the most frequent suggestions recorded when analyzing the interviews, where the question was the following: what are the strategies for continuing to develop healthcare professionals that provide sustainable quality healthcare to South Africans with the advent of the NHI? The answer we are proposing is the implementation of the ecosystem in Figure 4.

This implementation of the strategies requires the digital ecosystem shown in Figure 4. The actual development is carried out by the academic and training institutions through a cloud of an ecosystem made of public telecommunication infrastructures and IoT-enabled devices, where a healthcare professional, in accordance with the registration program, accesses online classes using healthcare institution credentials (Figure 4). For the population selected by the Department of Health, access is directly given to them by the Institution of Registration. The selection, according to the level, is based on a competitive entrance examination from which the best candidates are considered in an inclusive manner. Similar fairness is applied for the Healthcare Institution based on the training and development plan put in place. Registration is carried out according to the academic and training institutions’ offerings based on predetermined programs (see Figure 5).

### 5.1. Strategy 1: E-Learning/Online Learning

The use of e- and online learning in developing healthcare professionals to improve quality healthcare delivery is crucial for better time management at the workplace. The e- and online learning proposed here is computer-based telecommunication technology for medical training, learning, and assessment. It is a complete learning and/or training medical experience offered by academic institutions through their learning platforms. Blackboard and Moodle as learning tools have been famously used in South Africa and can still be used for this purpose. Participants are connected via sensor-based IoT devices and wireless 5G mobile access through their mobiles. The university develops standard learning programs in line with the curricula. Since practice is required, internships always follow each program, as per negotiations with the Department of Health for free-agent learners. At the end of a particular program, a certificate is issued as a record of the training or curricula completed.

Learners access the learning platform via the Internet and can also access libraries of health research evolution. Learners have the possibility to share their experiences through an AI assistant ChatBot, necessary for the learners’ experience evaluation. This evaluation, offered by the ChatBot, can be achieved through feedback that offers a Learner Satisfaction Score (LSS) for learner’s satisfaction and a Net Promoter Score (NPS) that records an individual’s performance for knowledge retention and continuous evaluation. When not required to carry out learning tasks by themselves, learners can follow live practice sessions from the hospital via Internet of Things (IoT) devices that display the required information on their computer, laptop, or mobile through the Internet.

For example, for a scan conducted at the hospital (Figure 6), learners can have the same image on their respective screens as the specialist conducting it via IoT, follow the analysis of the scan, and engage with the specialist for better comprehension.

Specialists can also improve themselves via expert discussion forums and the literature provided by the research institutions via the libraries (see Figure 4). An example can be a sensitive surgery operation; three or four experts can exchange their knowledge on the matter prior to carrying on with the proper operation.

### 5.2. Strategy 2: Tri-Partnership

For the excellent implementation of the CPD, active collaboration among the health institution, the academic or training institution, and the state institutions is required. This partnership is crucial: the state defines curricula, which are further developed by the university, and presented to the healthcare institution and the population in large. The university develops tailored-made programs that enhance skills among healthcare professionals. While health institution staff members get registered through an internal fair assessment, the population can only access the learning/training program through a competitive entrance examination based on excellence and transformation predispositions organized by the State Department. People from poor backgrounds access the NSFAS, high-income learners afford half (for example) of the fees, and healthcare-established professionals pay the totality, assuming the support of their institution. The State Department defines the medical standards and protocol, as well as the prerequisites to be eligible for learning, which are further applied by healthcare institutions. Healthcare institutions develop their staff but can also source talent from academic/training institutions, as well as from job-placement institutions, provided that the candidates fulfill the requirements. Job placements are mentioned here since there will be demand from smaller healthcare institutions like private practices and continuous replacement in bigger institutions.

### 5.3. Transformation and Outreach

Today, the South African social landscape needs transformation to achieve social cohesion, and one of the aspects to be considered after sports is education, and one sector to express education in terms of transformation is the healthcare system. The better the sector performs, the better society’s cohesion, and the more it disintegrates racial perceptions. Therefore, the outreach of the healthcare professional’s development, whether within healthcare institution or not, should be entirely embraced without excluding the merit of all races as well as all layers of society. The notion of merit should be redefined accordingly at the competitive entrance examination level. The reason being that a learner from an urban area and their rural counterpart cannot be assessed on the same basis, due to the simple fact that when one undertook full science classes, the other was only maybe occasionally, or not at all, undertaking science lessons. The State Department outreaches individuals through media channels; willing or encouraged candidates apply to participate in the competitive examination of their level. The admitted candidates are then selected for different programs based on their marks, considering also choice and transformation requirements. Though there is no measurement tool available to indicate transformation, but the social cohesion and general acceptance of one another can help survey whether there is transformation success.

### 5.4. Strategy 4: Programs

Programs constitute the nerve of the healthcare professional’s continuous development. Although specialists can improve their performance over experts’ discussion forums, general practitioners can use the same avenue or go back to upgrade to a specialist level. Some provisions are already offered in South Africa, as shown in [37]: When health practitioners who are actively practicing in South Africa attend an accredited professional or academic meeting or activity abroad, it will be recognized for CPD purposes. However, to assist these healthcare practitioners, there is a compelling aspect of tackling multiple crucial training needs within the organization such as “on demand” training encompassing the meticulous patient, maintenance of health records, cultivation of centric care approaches, optimization of supply chain logistics, facilitation of efficient knowledge management practices, and establishment of sound corporate governance principles labeled herein as administration (see Figure 5). Since CPD’s purpose is to assist health practitioners to maintain and acquire new and updated levels of knowledge, to gain skills and ethical attitudes that bring measurable benefits in professional practice, this program integrates technical skills development, life and soft skill development, internship programs, capacity building, and experiential learning into the curricula based on centric care and health management.

#### 5.4.1. Centric Care

Centric care refers to treating patients with dignity and respect and to involving them in all that is happening regarding their health. The assistance of healthcare practitioners is better represented with the following programs: emergency medical services, hospital- and clinic-based workforce-tailored healthcare professional training, healthcare management programs, and broader integrated training programs including home care and management support.

##### Emergency Medical Services

The program on emergency medical services focuses on technicians and includes the following:-Emergency medical technicians;-Life skills and life coaching.

The goal of this program is to produce competent emergency medical technicians who can serve the community with basic life support care using various EMS equipment. This goal can be enhanced with life coaching, mentoring, and experiential learning opportunities during internships.

##### Internal Healthcare Professional Training

Internal training is hospital- or clinic-based training, organized by the healthcare institution to upskill its employees. Several programs can be implemented:

Unskilled training programs: Programs intended to start from the basic skill level and advance up to a nursing assistant level and even to a qualified nurse level. Qualified nurse, however, require the curricula to be considered. This program seeks to expand skills among workers:-Remediation program: it is an individualized learner program that aims to fill gaps or disparities from a prior training or learning program. It is targeted support for struggling learners that reinforces foundational skills, addresses specific learner’s challenges, and enhances comprehension.-Career mapping program: it is a hospital-based training program that targets a specific position through related training.

##### The Curricula for Professional Programs

-Certified Nursing Program: The certified nursing program is conducted through curricula, during a three-year program.-Emphasized Laboratory Technician Training: This training, which initially gives basic laboratory skills to beginners, is reinforced through curricula with advanced education for entry level science labs. The advanced training stage is designed to advance skills and knowledge in basic mathematics, communication, and bio/lab-based professional development.

##### External Healthcare Professional Training

This level concerns all senior certificated individuals with good science understanding who failed to access tertiary education, but are willing to be trained as such. They are selected according to their performance. Primarily, the curricula here are made of institutional general knowledge dispensed at a university or college.

##### Health Management

Heath management includes social and environmental workers important in counseling patients living with dreadful disease or in a terminal phase and in directing waste and environmental issues, respectively. Their curricula are for academically oriented external candidates.

#### 5.4.2. Healthcare Management

It is important to have specialists in managing healthcare systems, particularly for the following.

##### Knowledge Management

Knowledge management helps identify, organize, store, and foster good practices by disseminating information within healthcare institutions. It is an emerging technology that manages the knowledge of an institution by structuring it for effective and efficient usage by employees and teams. This structuration includes the following, among others: information regarding revenue expansion strategies, recruitment best practices, winning bids for special employee skills in certain tasks, internal IT disciplines, and legal team strategies.

##### Supply Chain Operation

The supply chain is a set of activities required to take a product or service from conception, through the different steps of its production, a combination of its physical trans-formation and inputs of various producer services, and up to its delivery. This program helps analyze the benefits of an integrated supply chain and the requirements of inventory management. It focuses on quality improvements and innovation in managing the supply chain in terms of challenges and risk management.

##### Financial Management and Good Governance

Financial management is a crucial part of strategic planning and managing institution finances for better alignment of finances and goals. The program helps professionals to monitor, control, protect, and report the institution’s financials resources.

Good governance refers to the institutional policies and processes and outcomes that are necessary to achieve healthcare delivery.

## 6. Strategy Evaluation

The strategies are defined in an ecosystem where technological means are implemented to connect the people and the processes involved. Therefore, we evaluated these strategies within this environment, as shown in Figure 7.

### 6.1. Cost-Effectiveness Evaluation

There are several points gained in applying our healthcare professional’s development strategies, and two ways to evaluate the cost-effectiveness are effectiveness in offering training and effectiveness in harvesting from the training.

1.Efficiency

Efficient programs, according to current and future needs, whether for administration or medicine applications, social workers or environmentalists, or live hospital procedures as a class, guarantee quality training that allows learners to be more efficient once in the organization as workers. Healthcare institutions, compared to the SANC system, employ several skillful individuals who will probably improve resource management and quality healthcare delivery, which in turn will increase profitability.

Another aspect of evaluating efficiency is during training. The flexibility of a training timetable must be such that learners can work while attending training, which means that they can still produce benefits while gaining skills, hence improving efficiency during training.

2.Speed

With forum discussions happening after training within the ecosystem, solutions for big medical issues can be found faster compared to with the SANC and AHPCSA’s CPD, where practitioners work in isolation, exposing them to mistakes or the misinterpretation of certain outcomes.

During training, learners undergo tests instead of the SANC’s activities. Instead of reaching a certain number of points that give right to a certain professional practice, healthcare institutions have a faster, skilled professional ready to operate. This allows job position vacancies to fill in faster. The time process can even be shorter when the tri-partnership implements a communication scheme with job-placement entities.

This then speeds up the earlier care process of the patients, adding efficiency in the process and improving quality in delivery.

3.Quality

We cannot argue about quality healthcare delivery, which was one of the main points for suggesting an alternative in developing professionals. Applying this strategy continuously and timelessly will have, as a logical consequence, the achievement of sustainability, which comes with quality, speed, and costs as part of delivering healthcare services, while keeping the services and access constantly effective (Figure 8). Such a continuous and timeless program ensures the retention of workers, and the hiring of new talented workers. This allows us to post professionals in underserved areas like rural areas, permitting quality and sustainability extension.

4.Cost

Surely, cost containment, with skilled individuals available, should be guaranteed. Cost reduction is the logical consequence of investing in humans. The retention of staff through their training and development is an asset for the healthcare institution, since it will absolutely lead to less recruiting interviews, which are costly. Also, as the SDG requires a continuous investment of healthcare professionals by the state, provisions can be made to offer cost-effective training.

### 6.2. Financial Impact

The feasibility of such healthcare development requires proper financial funding. Though this paper does not focus on monetized aspects, it is at least important to express the financial impact.

Potential Challenges

Training Funding: It is the first challenge one can face. We must mention the NSFAS and healthcare institution funding as customers, and funding as pain points. The government, through the NSFAS and the healthcare institutions, may find it necessary, important, or indispensable to finance these strategies, which are necessary for the health and safety of the people, but this comes with an increase in expenses.

Start-Up: The biggest financial challenge is the start-up or up-front finances required to start with the pessimistic economy experienced these days. However, we believe that with political will, one will obtain the necessary fundings

Financial Harvest

Despite the challenges above, financial expenses are smaller compared to what these expenses will bring:Improve procurement and system inventory: given the resources needed to carry out procurement management and to conduct inventory in the healthcare system management, there will be efficiency in institution expenditures on healthcare purchases, which will improve the operations of the healthcare system.Improve financial resource management: having well-managed procurement and a proper follow-up of the healthcare institution’s financial state through inventory, and having gained good governance and financial management, one expects positive financial results that could foster further investments.Staff retention is an asset for the healthcare system in terms of benefits, since there is no need to recruit new staff, saving on the expenses required for hiring.

### 6.3. Challenges in the Implementation of the Proposed Strategies

There are several challenges when implementing the identified strategies that are confined in Figure 3, particularly when it comes to implementing them in an ecosystem:Transforming individuals into healthcare professionals will probably receive some skepticism from a minority of the population, particularly those trained from correctional services. However, with the resilience of social workers regarding rehabilitation, one can yield good people.The government’s expenses might push authorities to increase tax revenue. However, there are ways of financing the HI and the CPD simultaneously. Instead of increasing taxes uniformly, the government must reduce the tax burden of healthcare institutions, but at the same time ensure that the money from the tax reduction goes into the training and development plan.There is also the problem of mobile network access, since some areas are not properly covered by a mobile network, or perhaps a remote learner lacks data to access the network. However, potential deals can be set with different operators to provide free connections to learners.

## 7. Discussion

With cost-effectiveness being more about comparisons, the strategies proposed in this paper yield faster results compared to the SANC and AHPCSA’s CPD in terms of trained and developed employees since it is an embedded system. During their training, learners can access some live hospital sessions, making them faster and more effective, compared to the current system, which necessitates physical presence. Also, the inclusion of administration in the curricula gives them an edge to be more efficient once in the field. This yields more efficient and effective staff in a shorter period, which means reduced expenses, showing the cost-effectiveness required for the defined strategies. Furthermore, we introduced within the digital ecosystem a tool to evaluate the performance of the system through the learner’s experience. Learners’ and customers’ feedback are obtained via an AI-based survey conducted by an AI assistant ChatBot. The ChatBot can provide us with two indicators: an LSS to evaluate learner experience and NPS to evaluate the learner’s continuous knowledge. Both tools, the LSS and NPS, are identical to the Customer Satisfaction Score and Net Promoter Score used, respectively, in a digital economy. It is important to have feedback for the learner’s experience, since it allows for improvements and their career development. As for the NPS, it allows continuing development evaluation, by marking the score learners or trainees obtained through the questions and by recording each individual’s performance, replacing the controversial bias points previously used. This is new compared to the current scholarship systems that do not consider learner pain points in the system [42,43]. When compared to the current scholarship e-learning systems, most of them offer annual programs compared to the on-demand programs we are proposing, offering learner’s experience evaluation but not continuous knowledge evaluation. This places our digital ecosystem as a perfect tool for future healthcare professional development.

## 8. Conclusions

In this paper, we suggested strategies for continuously developing healthcare professionals. These strategies are derived from the analysis outcomes of a qualitative cost-effectiveness analysis conducted. Our contribution is a digital ecosystem for healthcare professionals’ development. This ecosystem is a digital platform that offers e- and online learning, structured in such a way that on-demand programs and live hospital procedures are watched during practice sessions, as well as an AI assistant such as a ChatBot for evaluating the learners’ experience and knowledge. We showed the cost-effectiveness of the system based on indicators such as efficiency, quality, speed, and cost, evaluated during and after training. We found that with continuous training, the system provides sustainable quality healthcare delivery. With quality healthcare delivery guaranteed, and with the continuous knowledge evaluation embedded, this supports the implementation of our development strategies in the context of the future, that is, as a digital ecosystem that, if used continuously for healthcare professionals’ development, will yield sustainable quality healthcare delivery, which is the main goal of this study. We also discussed challenges that come with such a solution, like network access and start-up finances, which might limit or delay the implementation in terms of resources, but there are several offerings that make our ecosystem a futurist tool for healthcare professionals’ development.

## Figures and Tables

**Figure 1 healthcare-13-00036-f001:**
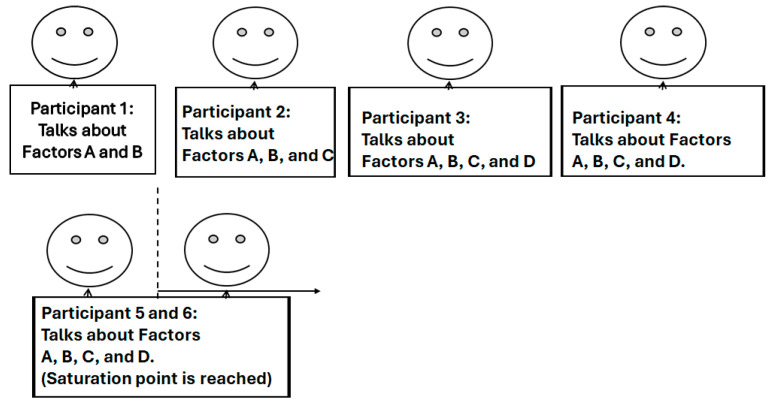
Conceptual framework and sampling technique for qualitative data.

**Figure 2 healthcare-13-00036-f002:**
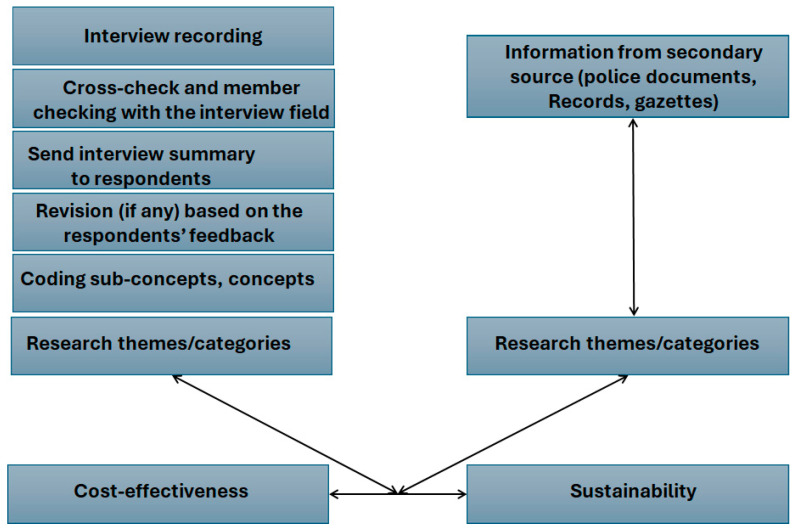
Steps for qualitative data analysis.

**Figure 3 healthcare-13-00036-f003:**
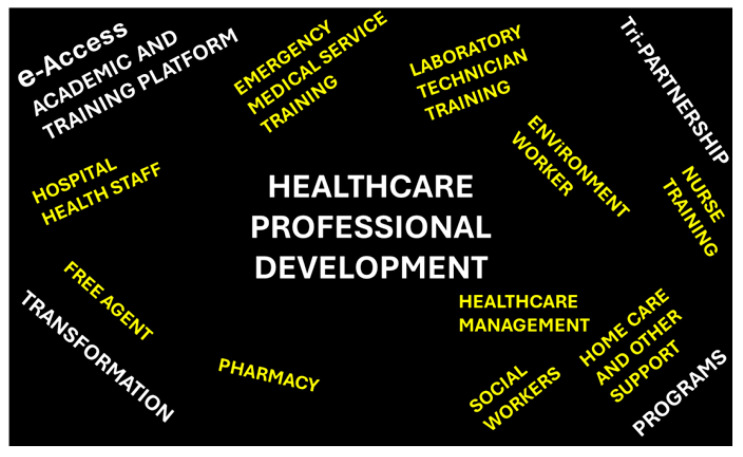
Strategies to improve the South African healthcare system.

**Figure 4 healthcare-13-00036-f004:**
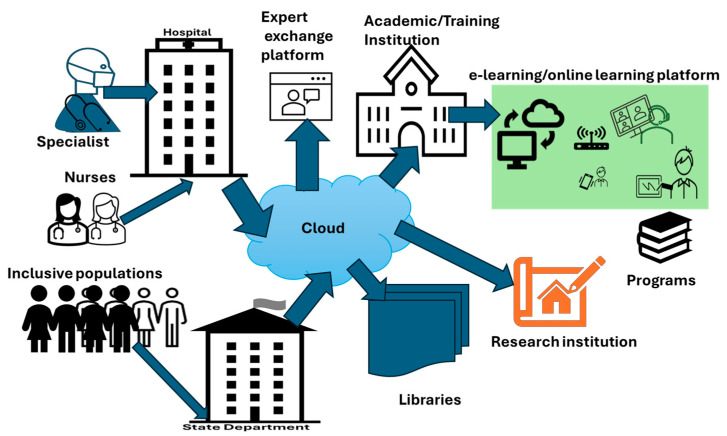
Digital ecosystem for professional development.

**Figure 5 healthcare-13-00036-f005:**
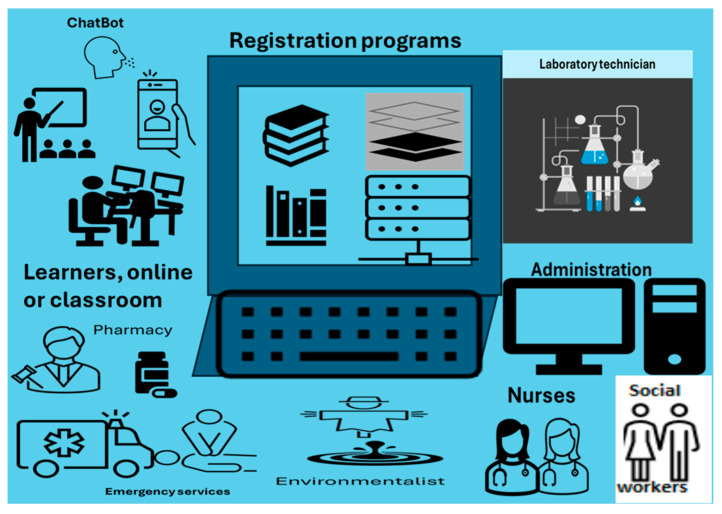
Learning and training development offerings.

**Figure 6 healthcare-13-00036-f006:**
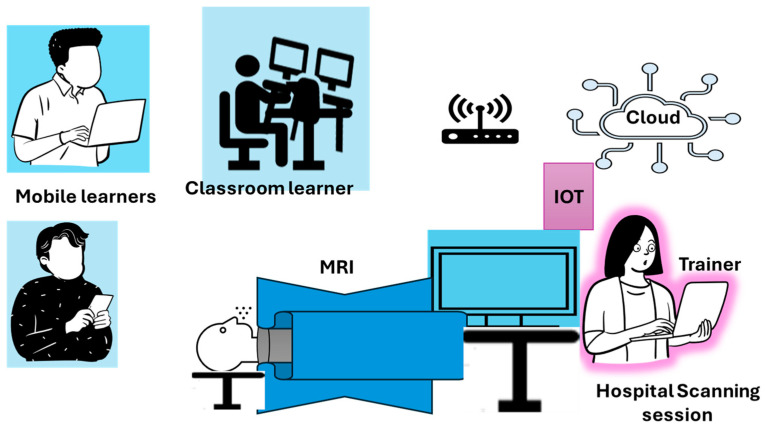
Learners during a scan practice session.

**Figure 7 healthcare-13-00036-f007:**
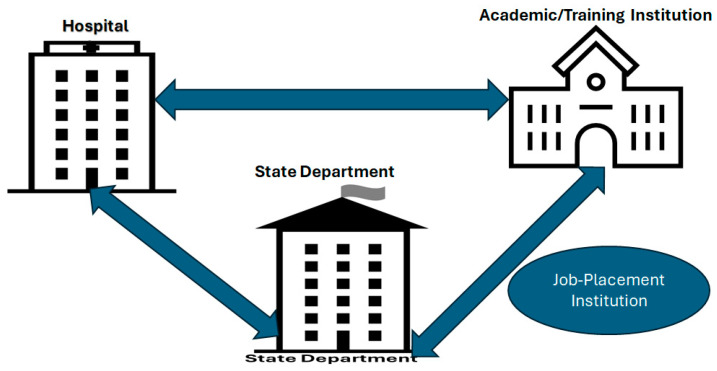
Tri-partnership.

**Figure 8 healthcare-13-00036-f008:**
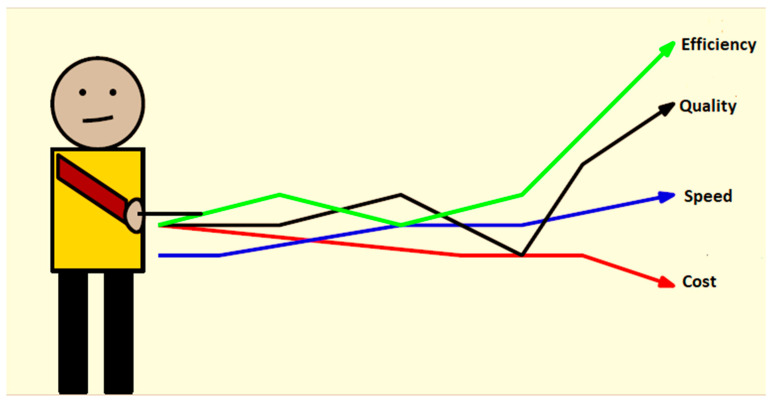
Results of strategies for cost-effectiveness analysis.

**Table 1 healthcare-13-00036-t001:** Numbers of healthcare professionals registered in South Africa [31].

Health Worker Categories	Number
General medical practitioners	29,311
Medical specialists	14,192
Dental practitioners	6155
Dental therapists	661
Professional nurses	140,598
Enrolled nurses	73,558
Nursing assistants	73,302
Pharmacists	14,412
Occupational therapists	4792
Physiotherapists	7183
Psychologists	8415
Radiographers	8072
Environmental health practitioners	3585
Clinical associates	577

**Table 2 healthcare-13-00036-t002:** Sample and characteristics profile (source data: interviews and surveys).

Category	Characteristics	No. of Samples	Percentage	Valid %
General practitioners	Private Practice	5	10	100
	Public Hospital	5	10	100
Nurses	Private Hospital	5	10	100
	Public Hospital	5	10	100
Social workers	Hospital Counseling	5	10	100
	Office Counseling	5	10	100
Emergency services	Private Ambulance	5	10	100
	Municipal Ambulance	5	10	100
Pharmacists	Pharmacy, Dispensary	5	10	100
	Ordinary Pharmacy	5	10	100

**Table 3 healthcare-13-00036-t003:** Interview questions.

No	Questions
1	What mode of healthcare professional development strategies is effective in improving the quality of healthcare delivery?
2	What mode of healthcare professional development strategies is cost effective for quality healthcare delivery?
3	What mode of healthcare professional development funding is effective for healthcare institutions?
4	What are some of the challenges or obstacles you faced with prospects of healthcare professionals’ development?
5	What healthcare professionals’ development strategies have you used to improve institution profitability and quality healthcare delivery?
6	What healthcare professionals’ development strategies have you used to improve institution profitability and quality healthcare delivery that failed?
7	In your experience, are there any healthcare professionals’ development strategies more effective in improving quality and profitability?
8	Is there anything else you would like to share about healthcare professionals’ development strategies to improve quality and profitability?

## Data Availability

Data will be made available as the need arises.

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
