# Peer review of "Strategies for Cost-Effectiveness in Sustainable Quality Healthcare Delivery in Emerging Economies: The Case of Healthcare Professionals Development in South Africa"

_healthcare, 2024, doi:10.3390/healthcare13010036_

Round 1
Reviewer 1 Report
Comments and Suggestions for Authors
Major revision
Review Report
Health care services’ improvements become a key policy agenda under the sustainable development goals (SDGs) in any country of today. Out of the package of the entire service, the services from the professionals are very crucial. The present paper has aimed at proposing some policies towards improvements of these services, ceteris paribus, for the emerging economies especially the rural belt of South Africa. The study has used statistical and econometric tools for reaching the results and their analyses. The results are expected as there are so many studies in the similar areas. It has some degrees of policy implications so far as the importance of improving healthcare services from the professionals, in particular, in order to attain the sustainable development. However, the study has some lacunae in some of the areas as mentioned below and the rectifications/revisions of which may be leading to a good research outcome.
1. The Abstract should start with a background of the problem, specific objective/s, data & methodology used, and the derived major results. All the sequences are missing in the paper and it is suggested to thorough revision of the abstract.
2. The study has a brief introduction to the subject area. It starts with the issues related to health care service status and the ways of improving it. But it did not call out anything about why such issue becomes important. Whether the causes behind poor health conditions could be mentioned. The authors are suggested to mention, for better organization and background of the study, the roles of urbanization, non agricultural activities, pollution and health cost relationships etc, (like is pollution making cost to health, etc) and restructure the Introduction part.
3. It is not clear about the motivations behind the study? All have no clear explanation; it is suggested to incorporate the same. Further, the study should categorically mention the Research Gaps to justify its contributions to the scientific literature as it used the data and results of so many other authors.
4. It is a natural question whether the health care services were poor in the study area in particular, and the country, S Africa, in general. No statistical evidences are there, neither any secondary data presentations nor the authors’ own exercise. Thus, it is a vague effort in chasing behind the objectives under the study.
5. The survey method is commendable as it considered the revised opinions of the respondents, but it is not observed which empirical methodologies were followed by the authors. Simply by taking the data and putting some intuitions from the part of the authors could not justify any policy strategies. Thus, it is suggested to present the detailed methodology and computed results for the further development and concreteness of the efforts.
6. How the cost-effective analysis is done not clear, as to do this, the authors have to make comparisons between all costs and benefits in suggesting such policies (CPD). It is completely missing.
7. The study did not present any theoretical basics of the interrelationships between the two key indicators without having any model. It is suggested to form a basic model with at least linking the two variables in equational structure to give a break to the monotonicity in the presentations.
8. The author should also discuss the policy formulation in further detail and should be all inclusively analysed in the Discussion part.
9. There are certain typos such as cost-effective… in line 67, etc. which are to be removed.
Major revisions recommended………………..
Comments on the Quality of English Languageminor editing
Author Response
Dear Reviewer 1,
- Comment: The Abstract should start with a background of the problem, specific objective/s, data & methodology used, and the derived major results. All the sequences are missing in the paper and it is suggested to thorough revision of the abstract.
Response: The abstract has been revised accordingly
- Comment: The study has a brief introduction to the subject area. It starts with the issues related to health care service status and the ways of improving it. But it did not call out anything about why such issue becomes important. Whether the causes behind poor health conditions could be mentioned. The authors are suggested to mention, for better organization and background of the study, the roles of urbanization, non agricultural activities, pollution and health cost relationships etc, (like is pollution making cost to health, etc) and restructure the Introduction part.
Response: The importance of our proposal is added in the introduction as well as the overall improved background (Highlighted parts)
- Comment: It is not clear about the motivations behind the study? All have no clear explanation; it is suggested to incorporate the same. Further, the study should categorically mention the Research Gaps to justify its contributions to the scientific literature as it used the data and results of so many other authors.
Response: The contribution has been modified accordingly in the introduction as highlighted before the last paragraph.
- Comment: It is a natural question whether the health care services were poor in the study area in particular, and the country, S Africa, in general. No statistical evidences are there, neither any secondary data presentations nor the authors’ own exercise. Thus, it is a vague effort in chasing behind the objectives under the study.
Response: Section 3 related to healthcare system in South Africa has been modified accordingly
- Comment: The survey method is commendable as it considered the revised opinions of the respondents, but it is not observed which empirical methodologies were followed by the authors. Simply by taking the data and putting some intuitions from the part of the authors could not justify any policy strategies. Thus, it is suggested to present the detailed methodology and computed results for the further development and concreteness of the efforts.
Response: Section 4.3 Data collection method has been corrected accordingly
- Comment: How the cost-effective analysis is done not clear, as to do this, the authors have to make comparisons between all costs and benefits in suggesting such policies (CPD). It is completely missing.
Response: Section 4.3 Data collection describes how cost-effectiveness is made used of.
- Comment: The study did not present any theoretical basics of the interrelationships between the two key indicators without having any model. It is suggested to form a basic model with at least linking the two variables in equational structure to give a break to the monotonicity in the presentations.
Response: Section 6 on Evaluation brings more on this in accordance with the definition given of cost-effectiveness.
- Comment: The author should also discuss the policy formulation in further detail and should be all inclusively analysed in the Discussion part.
Response. A discussion section 6.4 was added for the circumstance.
- Comment: There are certain typos such as cost-effective… in line 67, etc. which are to be removed.
Response. revised
Reviewer 2 Report
Comments and Suggestions for Authors
Thank you for the opportunity to review this manuscript. The manuscript presents an interesting topic; however, it needs major revision to make it publishable. Here are some suggestions to enhance its clarity, coherence, and impact:
Introduction
# Clarity and Conciseness: Simplify complex sentences and remove redundant phrases for better readability. For example, instead of “Advancements in technology have led to the profound transformation,” simply state, “Technological advancements have transformed…”
# Clarify ambiguous terms like "adaptive complex regulatory" and "const-effective" (likely intended to be “cost-effective”).
# Grammar and Syntax: Check for grammar and typographical errors, such as "const-effective" instead of "cost-effective" (line 67).
# Avoid using "encompasses a wide array of critical areas, including but not limited to," as it is verbose. Simply stating "includes" would be sufficient.
# Logical Flow: The structure in certain parts can be improved for logical flow. Consider reorganizing the Introduction to start with a broad context on healthcare advancements, then focus on South Africa’s National Healthcare Insurance (NHI), and finally introduce the study's focus on CPD.
# Terminology: Introduce and define key terms and acronyms like CPD (Continuing Professional Development) and CME (Continuing Medical Education) before using them.
# Avoid redundancy by not defining acronyms multiple times. For instance, CPD is introduced twice in separate parts.
# Adding more background on the importance of digital health and CPD across other health systems can strengthen the context.
# Include recent studies and examples of digital CPD platforms to add relevance and substantiate claims.
# Improve section transitions to make it easier for readers to follow the paper’s structure.
# Clarify the study’s primary research question or objective early in the Introduction. Instead of broadly stating it “explores strategies aiming to develop healthcare professionals,” specify the exact scope of the study and the unique contribution of your research.
# Outline the specific research methods, sampling, data analysis, and how cost-effectiveness is measured.
# Mention specific cost-effectiveness tools used and the rationale behind choosing those.
Literature Review
# Ensure Terminological Consistency: Terms like "CPD," "CME," "digital ecosystem," and "cost-effectiveness analysis" should be defined clearly and used consistently throughout the manuscript to avoid confusion. For example, CPD is defined differently in several sections; unify the definition early on.
# Improve Flow and Transitions in Literature Review: Streamline the literature review by logically grouping themes (e.g., technology’s role in CPD, cost-effectiveness in healthcare) and ensure smooth transitions between ideas. Consider synthesizing findings on CPD and healthcare technology advancements from different sources to build a cohesive argument.
# Expand on Cost-Effectiveness Analysis Framework: Provide a more detailed explanation of cost-effectiveness analysis in Section 4. Explain specific metrics or methods used in healthcare cost-effectiveness studies and why this approach is beneficial for resource allocation in South Africa.
# Strengthen Section on South Africa’s Healthcare System (Section 3): Expand on healthcare disparities due to socioeconomic and racial factors with additional data, if available, to reinforce the argument. Discuss how these disparities impact CPD opportunities and the feasibility of a national digital platform for CPD.
# Incorporate More Recent References: Ensure all references are recent and relevant to the South African context. Citing recent research on technology in healthcare and CPD effectiveness would strengthen the study’s contemporary relevance.
# Develop a Clear Methodology for Cost-Effectiveness Study: Outline the specific steps taken in the cost-effectiveness study conducted in South Africa. Detail the tools, methods, and sample demographics used to gather data, making it clear how findings were derived and analyzed.
# Structure the Literature Review to Emphasize Gaps and Study Contributions: Conclude the literature review with a section highlighting existing research gaps in CPD and digital solutions in South African healthcare. Specify how this study addresses these gaps, reinforcing its novelty and significance.
# In Section 4, provide detailed, actionable strategies for implementing a digital CPD system in South Africa. Consider discussing specific technologies, training modules, or partnerships with educational institutions.
# Improve Language and Grammar: Carefully edit the manuscript for grammar, clarity, and syntax. Ensure the language is formal, direct, and avoids redundancy. For instance, phrases like "undoubtfully has taken a path of its revolution" should be rephrased for clarity.
# Wherever possible, compare South Africa’s healthcare system and CPD practices with those of other countries, especially emerging economies, to provide context and reinforce the need for a tailored solution.
# Consider including tables or figures that outline the proposed CPD strategy, cost-effectiveness metrics, or healthcare disparities. Visual aids can enhance comprehension and make the study more engaging.
Methodology
# Clarify Research Objectives in Method Section: Clearly state the specific research objectives in Section 4.1 to link the qualitative case study approach to the broader goals of the study. This will provide context and help justify the choice of methods.
# Specify the Data Collection Methods and Justify Their Use: In Section 4.1, explain why surveys, observations, and interviews are suitable for exploring healthcare management practices and how each method contributes uniquely to the findings. Distinguish between the intended outcomes from surveys, observations, and focus groups.
# Detail Virtual Interview Platforms’ Relevance: In Section 4.1, briefly explain the decision to use multiple virtual platforms (Zoom, Teams, Google Meet) and telephone interviews. Highlight how this choice accommodates participant convenience or broadens access, especially in the healthcare setting.
# Provide Sampling Criteria Rationale: Expand on the rationale behind the three criteria for sample selection in Section 4.2 (demographic characteristics, facility size, geographical location). Clarify how each criterion relates to the study’s objectives in examining cost-effectiveness and quality healthcare delivery.
# Explain Saturation Point in Sampling Technique: Elaborate on reaching the "saturation point" in the sampling process. Provide more detail on how many participants were initially targeted, how many were ultimately interviewed, and what specific trends or repetitive responses led to the conclusion of data saturation.
# Improve the Sampling Technique Section’s Structure: Break down the sampling steps in Section 4.2 for clearer readability. For example, use subheadings or bullet points to delineate steps like initial participant selection, criteria for expansion, and reaching saturation.
# Strengthen the Data Collection Explanation: In Section 4.3, clarify the role of primary and secondary data sources and how they complement each other. Briefly explain the types of policy documents, records, and published materials reviewed and how these sources informed the study’s findings.
# Describe the Data Coding Process in Greater Detail: Expand the coding description in Section 4.3. Explain the coding framework, tools used, and specific examples of how raw data was transformed into sub-concepts and main concepts to arrive at research outcomes. This will enhance the reliability of the data analysis process.
# Explain Data Validation Techniques: Provide additional information on the process of sending summarized data back to participants for validation. Specify how feedback was incorporated, if participants made any corrections, and whether this step improved the accuracy and credibility of the data.
# Use a More Descriptive Figure Caption: The caption for Figure 1 (“Conceptual framework technique”) could be more descriptive, such as “Conceptual Framework and Sampling Technique for Qualitative Data Collection.” Make sure the figure itself visually represents the sampling and data coding processes clearly.
# Address Potential Limitations of the Methodology: Briefly discuss any limitations of the qualitative case study approach in Section 4, such as potential biases in participant selection or challenges related to virtual data collection.
# Correct minor grammatical errors (e.g., “makes use of” could be simplified to “uses” in Section 4.1). Review sentence structure for clarity and flow, particularly in longer sentences, to enhance readability.
# Provide References with More Recent Studies or Methods: Ensure that citations are recent and relevant to the South African healthcare context or qualitative methodology. Update any references that may be outdated or replace them with more recent sources on qualitative case studies in healthcare.
# Consolidate Redundant Information: Some sections, such as the mention of virtual platforms in Section 4.1, repeat the same point. Consolidate similar information to reduce redundancy, making the methodology section more concise.
Data Analysis
# Clarify Figure 2 Title and Caption: Change "Figure 2. Steps for Qualitative Data Analysis" to a more descriptive title, such as "Sequential Steps for Analyzing Qualitative Data in South African Healthcare Context." This provides clarity on the focus of the data analysis.
# Explain the approach used to analyze qualitative data (e.g., thematic analysis, grounded theory, or content analysis). Mention specific techniques for identifying patterns and themes from the data, especially in Section 4.4.
# Justify Each Outcome’s Relevance: Provide a brief justification for each of the four outcomes in terms of the study’s goals. For example, explain why e-learning (Outcome 1) or tri-partnerships (Outcome 2) are essential for improving the healthcare system in South Africa.
# Strengthen Connection Between Outcomes and Data Sources: Clearly state how each outcome emerged from different data sources (surveys, interviews, observations). This will enhance the transparency of how the data collection methods informed the results.
# Provide Participant Quotes for Key Outcomes: Integrate participant quotes or paraphrased responses for each outcome. For instance, a quote highlighting the importance of e-learning could substantiate Outcome 1, making the results more vivid and directly tied to participant perspectives.
# Add Examples for Key Suggestions: For Outcome 1, include specific examples of e-learning programs or training modules relevant to healthcare in South Africa. Similarly, for Outcome 4, provide examples of specific emergency medical services or healthcare management programs.
# Clarify “Transformation Outreach” Terminology: Define “Transformation Outreach” more explicitly in Outcome 3. Describe what “transformation” entails in the South African healthcare context, and how it can be practically implemented in training and hiring processes.
# Avoid vague terms like “it is also an occasion for the healthcare system to undergo transformation” (Outcome 3). Specify how transformation can be measured and tracked within healthcare training and hiring, and what specific changes it entails.
# In Outcome 2, explain the role of government health and education departments, academic institutions, and healthcare providers in more detail. Outline each party’s responsibilities in this partnership for a clearer understanding of the collaborative effort.
# Structure Outcomes More Clearly: Use subheadings or bullet points to make each outcome distinct and improve readability. For example, bold each outcome title (e.g., Outcome 1: E- and Online Learnings), then explain the findings below it.
# Correct grammatical errors, such as "the im-portance of having" (change to "the importance of providing") and "makes use of" (simplify to "uses"). Ensure consistent sentence structure for readability.
# Some details, especially within Outcomes 2 and 3, are repeated. Streamline these sections to avoid redundancy and focus on the main points.
# In Figure 3, visually represent how the strategies contribute to specific healthcare goals, such as cost-effectiveness, patient care quality, and inclusivity. Consider a flowchart to show how each strategy interlinks with the goals.
# Briefly discuss potential challenges for each outcome. For instance, mention the limitations of online learning accessibility (Outcome 1) or the potential difficulties in coordinating tri-partnerships (Outcome 2).
# Conclude Section 4.4 by linking the outcomes to potential policy recommendations for the South African healthcare system. This can underscore the study’s practical contributions.
# Provide relevant literature or case studies that back each outcome, particularly for e-learning in healthcare, tri-partnerships, and diversity/inclusivity in medical training.
Proposed Implementation of Continuous Professional Development Strategies
# Consider breaking down dense paragraphs into shorter, more digestible ones for easier readability.
# Ensure consistent terminology for similar ideas to avoid confusion, particularly around “digital platform ecosystem” and “e-learning.”
# Ensure all figures (e.g., Figures 4, 5, 6) are of high quality and adequately labeled for readability.
# Improve captions to provide brief explanations of what each figure represents without readers needing to refer to the text extensively.
#Add references to each figure in the text at appropriate points to guide readers.
# Standardize the structure used to describe each strategy (e.g., introduction, objectives, method, outcomes) for easier comparison.
# Expand on certain strategies, such as "Tri-partnership," by providing more concrete examples of implementation steps or best practices.
# Include references to relevant theories or studies to contextualize strategies (e.g., frameworks in continuous professional development, telemedicine).
# Discuss briefly any existing models for professional development that influenced your ecosystem approach.
# Clarity on Financial Feasibility: The "Financial Impact" section could benefit from a clearer breakdown of funding sources and potential financial challenges.
# If feasible, provide a rough estimate or comparative analysis of costs for implementing such a program, supported by secondary data or case examples.
# Specific Examples of Technology Usage: Expand on the use of technology, particularly IoT devices, in training sessions. Provide examples or case scenarios to demonstrate their effectiveness.
# Elaborate on how real-time data sharing through IoT improves learning outcomes in the practical sections.
# In the "Sustainable Quality Healthcare" section, explain how the program supports long-term sustainability in more detail, particularly regarding maintaining trained professionals in underserved areas.
# Discuss any expected impacts of the strategies on healthcare outcomes, patient satisfaction, and quality of care, supported by any preliminary or theoretical evidence.
# Simplify Technical Language: The language in certain sections (such as "supply chain operation" and "centric care") could be simplified for broader accessibility.
Avoid overuse of complex jargon; replace with simpler terms or explain terms briefly when used.
# In the "Strategies Evaluation" section, expand the criteria beyond cost-effectiveness to include metrics for evaluating learner satisfaction, knowledge retention, and long-term career development.
# Specify any tools or methods that could be used to measure these evaluation criteria, making the section actionable.
# Check for typos and grammatical errors (e.g., “mamngement” instead of “management”) throughout the document.
# Ensure sentence structures are correct and that each sentence conveys its point clearly and concisely.
Conclusion
# Clarify the Main Contributions: Clearly articulate the main contributions of the paper. Rather than stating that strategies were suggested, emphasize the innovative aspects or unique insights provided by these strategies in developing healthcare professionals.
# Reframe for Clarity and Precision: Simplify language to enhance clarity. For instance, rephrase “yielded as results, quality and sustainable healthcare delivery, efficiently with finance and other administrative tasks” to something more direct, such as “demonstrated cost-effective and sustainable improvements in healthcare delivery.”
# Address Typographical Errors: Correct typos and grammar errors, such as "te main goal" (should be "the main goal") and "might be delayed to coming out of eggs" (this metaphor is unclear and could be rephrased more directly).
# Revisit the Digital Ecosystem Context: Provide a more specific link between the proposed strategies and the concept of a digital ecosystem. Explain briefly how each strategy supports or aligns with a digital future in healthcare professional development.
# Expand on Limitations and Future Research: Expand on the limitations of the study, specifically regarding the administrative hurdles mentioned. Outline which specific areas need further investigation or development and why these are essential for implementing the digital ecosystem approach.
# Strengthen the Practical Implications: Emphasize the practical implications of the strategies for stakeholders, such as policymakers, healthcare managers, and educators. Summarize how these strategies might impact these groups in implementing a quality healthcare ecosystem.
# Provide a Brief Summary of Findings: Include a concise summary of key findings from the qualitative cost-effectiveness analysis. This will give readers a quick overview of how each strategy contributes to quality and sustainability in healthcare delivery.
# Reinforce the Vision for Future Implementation: Rather than suggesting delays, end on a forward-looking note. Highlight the potential for these strategies to transform healthcare education and training if adopted, despite administrative challenges.
# Remove Ambiguities in Language: Replace vague expressions such as “coming out of eggs” with clear, direct language. For example, “implementation may be delayed due to administrative steps that require further clarification.”
# Review the conclusion for fluency and logical flow. Ensure each sentence naturally connects to the next and that ideas build on one another to conclude the paper effectively.
General comments
# Please follow a typical academic research paper style with an Introduction, review of literature, methods, Data analysis, results, discussion, and conclusion.
# The language in certain sections could be simplified for broader accessibility.
# Check for typos and grammatical errors
By implementing these suggestions, you can enhance the clarity, depth, and impact of your manuscript..
Wish you all the best with this paper!
Comments on the Quality of English Language
The language in certain sections could be simplified for broader accessibility.
Check for typos and grammatical errors
Reviewer 3 Report
Comments and Suggestions for Authors
This is an interesting paper with a unique focus on the possibility for professional development in the health sector. There are some areas where the paper can be improved. These are as follows:
1. Introduction-the introduction speaks to the role of professional development but not enough on the factors affecting the health sector. It may be advisable that the authors contextualize the study within the scholarship of the field and bring the section on the South African context earlier. In doing this, more discussion on the achievements in the health sector for South Africa. Only the gaps or the negatives are presented here.
2. Methods-This section of the paper talks about the use of surveys, documentary analysis and interviews. However, there is no data in the reporting of the findings that speak to each of these. A revision is recommended for this section.
3. Findings-There is the use of interviews for this study. Yet, there is no use of quotations or verbatim statements as is the practice in qualitative research to demonstrate the richness of this method. It is advisable that these be included within the revision.
4. Discussion-the authors present a description of the findings but do not locate this within the broader scholarship and within the literature for the sector. It is recommended here that the findings be compared or situated within existing scholarship
Author Response
Dear Reviewer 3,
- Introduction-the introduction speaks to the role of professional development but not enough on the factors affecting the health sector. It may be advisable that the authors contextualize the study within the scholarship of the field and bring the section on the South African context earlier. In doing this, more discussion on the achievements in the health sector for South Africa. Only the gaps or the negatives are presented here.
Response: The introduction has been modified accordingly.
- Methods-This section of the paper talks about the use of surveys, documentary analysis and interviews. However, there is no data in the reporting of the findings that speak to each of these. A revision is recommended for this section.
Response: The entire section has been revisited
- Findings-There is the use of interviews for this study. Yet, there is no use of quotations or verbatim statements as is the practice in qualitative research to demonstrate the richness of this method. It is advisable that these be included within the revision.
Response: The quotes from some respondents are added accordingly
- Discussion-the authors present a description of the findings but do not locate this within the broader scholarship and within the literature for the sector. It is recommended here that the findings be compared or situated within existing scholarship
Response A Discussion section, Section 7 has been included and does the required broadening of our suggestion
Reviewer 4 Report
Comments and Suggestions for Authors
Review of “Strategies for Sustainable Quality Healthcare delivery in Emerging Economies: Case of Healthcare Professionals Development in South Africa”
Summary
This paper explores strategies for continuous professional development in healthcare institutions to ensure quality and sustainable healthcare delivery, focusing on South Africa. The authors propose a digital ecosystem for professional development and evaluate its cost-effectiveness.
Primary Comments
The topic is both interesting and important, and the paper is well-organized. My main comment is outlined below.
As indicated in line 192, "All five participants," the study relies on only five survey responses, with some results based on even fewer. For instance, Outcome 3 is derived from just one participant. This severely limits the generalizability and validity of the findings, and I recommend that the authors increase the sample size.
Additionally, the methods and reasoning leading to some results and conclusions are unclear. It’s difficult to understand how certain outcomes can be achieved. The authors should make a stronger effort to clarify their conclusions and the processes used to reach them.
Secondary Comments
Figures 1 and 2 in the paper are blurry and hard to read.
The content in Figure 3 is disorganized and somewhat confusing, which is also true for some of the other figures.
On page 5, the text states, "All these steps are summarized in Figure 1," but it seems this should actually refer to Figure 2.
Additionally, there are several grammatical errors throughout the paper that need to be corrected.
Comments on the Quality of English LanguageThere are several grammatical errors throughout the paper that need to be corrected.
Author Response
Dear Reviewer 4,
Primary Comments: The topic is both interesting and important, and the paper is well-organized. My main comment is outlined below.
As indicated in line 192, "All five participants," the study relies on only five survey responses, with some results based on even fewer. For instance, Outcome 3 is derived from just one participant. This severely limits the generalizability and validity of the findings, and I recommend that the authors increase the sample size.
Response: The entire sample size has been described accordingly
Additionally, the methods and reasoning leading to some results and conclusions are unclear. It’s difficult to understand how certain outcomes can be achieved. The authors should make a stronger effort to clarify their conclusions and the processes used to reach them.
Response: The methods used to reach the results presented have been clarified
Secondary Comments: Figures 1 and 2 in the paper are blurry and hard to read.
Response: Figure adjusted accordingly
The content in Figure 3 is disorganized and somewhat confusing, which is also true for some of the other figures.
Response: Figure represents strategies in the four conners wrapping the training programs
On page 5, the text states, "All these steps are summarized in Figure 1," but it seems this should actually refer to Figure 2.
Response: Correction made
Additionally, there are several grammatical errors throughout the paper that need to be corrected.
Response: Concern attended
Round 2
Reviewer 1 Report
Comments and Suggestions for Authors
Good revision
Author Response
Reviewer's comment: Good revision
Reviewer 2 Report
Comments and Suggestions for Authors
Thank you for the revisions. I recommend to accept the manuscript.
Author Response
Reviewer's comment: Thank you for the revisions. I recommend to accept the manuscript.
Reviewer 3 Report
Comments and Suggestions for Authors
Dear Authors,
Thank you for attending to the suggested revisions for this paper. The contextual description has been well extended also to help the reader. There are areas where some of the suggestions however remain outstanding. These would include:
1. Methodology. There is reference to a qualitative case study with surveys, secondary data analysis and interviews. However, none of the data from these data collection points are presented. The themes are listed and slightly discussed but without the use of quotations, which presents the richness of qualitative research. A further suggestion is to select three of the major themes for the study and to provide evidence/data/quotations that speak to the themes.
Discussion-This can be strengthened with an engagement of the findings, vis-a-vis those within the field to locate the contributions of the paper.
Additionally, I would be useful to discuss more qualitative studies within examinations of the literature that tells the reader more about the experiences of persons who have used or been trained with CPD. This would strengthen the qualitative focus of the study.
Author Response
Dear Reviewer 3,
1. Methodology. There is reference to a qualitative case study with surveys, secondary data analysis and interviews. However, none of the data from these data collection points are presented. The themes are listed and slightly discussed but without the use of quotations, which presents the richness of qualitative research. A further suggestion is to select three of the major themes for the study and to provide evidence/data/quotations that speak to the themes.
Response: Table 2 has been added for survey and interviews, as well as reference [47] is added to extend [46] to present the primary and secondary data points.
The “quotes” are added accordingly in 4.4 Data analysis section,
2. Discussion-This can be strengthened with an engagement of the findings, vis-a-vis those within the field to locate the contributions of the paper.
Response: We added quotes in Data Analysis section, and important elements in Section 3 as well as in the Discussions section, that captivate readers, allow them to be figuring out the current system through experienced professionals quotes, and visualizing our contribution.
3. Additionally, I would be useful to discuss more qualitative studies within examinations of the literature that tells the reader more about the experiences of persons who have used or been trained with CPD. This would strengthen the qualitative focus of the study.
Response: See above response
Reviewer 4 Report
Comments and Suggestions for Authors
I do not have other comments.
Author Response
Reviewer's comment: I do not have other comments.